# Burden of diabetic ketoacidosis and its predictors among diabetic patients in Ethiopia: Systematic review and meta-analysis

Sefineh Fenta Feleke[1]*, Anteneh Mengist Dessie[2], Zenebe Daniel Getachew[3], Fasikaw Kebede Bizuneh[1], Atitegeb Abera Kidie[1], Berihun Mulu Yayeh[1], Birtukan Gizachew Ayal[1], Natnael Amare Tesfa[3]

1 Department of Public Health, College of Health Sciences, Woldia University, Woldia, Ethiopia,
2 Department of Public Health, College of Health Sciences, Debre Tabor University, Debre Tabor, Ethiopia,
3 School of Medicine, College of Health Sciences, Woldia University, Woldia, Ethiopia

* fentasefineh21@gmail.com

**Data Availability Statement:** All relevant data are within the manuscript and its Supporting Information files.

## Abstract

### Background

Multiple studies across Ethiopia have investigated the occurrence of DKA, showing significant variations and conflicting findings. This systematic review and meta-analysis seek to consolidate the overall prevalence of diabetic ketoacidosis and its associated factors in the Ethiopian context.

### Methods

The study adhered to the Preferred Reporting Items for Systematic Reviews and Meta-Analysis Protocols (PRISMA-P) guidelines. Data was collected from PubMed/MEDLINE, Science direct, Google Scholar, and gray literature sources. Microsoft Excel was used for data extraction and summary, while the analysis was performed with R software version 4.3.2. The overall pooled prevalence of diabetic ketoacidosis and its components was estimated using a random effects model. Publication bias was assessed both graphically, using funnel plots, and statistically, with tests such as Egger's regression test. Subgroup analysis were carried out to minimize random variations in the estimates from the primary studies.

### Result

The pooled estimated prevalence of diabetic ketoacidosis among diabetic patients in Ethiopia was 46% (95% CI: 36, 57; I2 = 100%, P≤0.001). Medication discontinuations (AOR = 1.30, 95 CI 1.20, 1.64), presence of comorbidity (AOR = 1.53, 95 CI 1.10, 2.20) and presence of infection (AOR = 1.62, 95 CI 1.31, 1.98) had an association with diabetic ketoacidosis among diabetic patients.

### Conclusions

Medication discontinuations, comorbidity, and infection are individual contributors to diabetic ketoacidosis in diabetic patients. Implementing initiatives to enhance medication

**Funding:** The author(s) received no specific funding for this work.

**Competing interests:** The authors have declared that no competing interests exist.

adherence and establish comprehensive diabetes management programs covering glycemic control, comorbidities, and infection management can effectively address these factors.

## Introduction

Diabetes Mellitus comprises a collection of prevalent metabolic disorders marked by persistent high blood sugar levels, involving substantial dysfunction in the processing of carbohydrates, lipids, and proteins [1]. This condition stands as a primary contributor to end-stage renal disease, amputations of the lower extremities due to trauma, adult blindness, and an increased susceptibility to cardiovascular diseases [2]. Diabetic ketoacidosis (DKA) is a severe and potentially life-threatening complication arising from diabetes. It is most prevalent in individuals diagnosed with type 1 diabetes, but it can also occur in those with type 2 diabetes [3].

While diabetic ketoacidosis is commonly associated with type 1 diabetes, instances have been documented in individuals with type 2 diabetes. Despite type 1 diabetes patients typically experiencing more severe acidosis, individuals with type 2 diabetes may also need acidosis treatment, as they can develop diabetic ketoacidosis [4]. Factors like non-compliance, new-onset diabetes, infections (particularly pneumonia and urinary tract infections), as well as conditions like alcohol abuse, trauma, pulmonary embolism, myocardial infarction, and certain medications, including corticosteroids and antipsychotic drugs, may precipitate DKA [5].

Diabetic ketoacidosis rates range from 0 to 56 cases per 1000 person-years in diverse studies. It is more prevalent in women and non-white populations, and its incidence is higher in those using injectable insulin compared to subcutaneous insulin infusion pumps [6]. Diabetic ketoacidosis stands as the primary cause of death in children and young adults, constituting approximately 50% of fatalities, with half occurring in individuals under 24 years old [7]. The treatment of DKA consumes significant resources, contributing to an estimated annual total cost of $2.4 billion [8].

In Africa, around 19.8 million people (4.9%) had diabetes in 2013. In Ethiopia, the World Health Organization (WHO) projected 800,000 diabetes cases in 2000, expecting an increase to 1.8 million by 2030. Studies from 1970 to 2011 in Ethiopia showed a diabetes prevalence of about 2%, rising to over 5% in individuals aged 40 and older in certain settings. A recent nationwide WHO Steps survey in Ethiopia, with 2153 participants, reported a 6.5% diabetes prevalence [9].

In low-income nations like Ethiopia, healthcare challenges arise from limited resources and insufficient insulin supply in hospitals, affecting the thorough investigation of patients. This, combined with sub-optimal nutrition in economically constrained individuals, especially during their first hospital visit for diabetes, may lead to unfavorable outcomes when implementing intensive insulin therapy for diabetic ketoacidosis, mimicking re-feeding syndrome [10].

Despite the existence of a systematic review and meta-analysis on diabetic ketoacidosis (DKA) conducted in Ethiopia [11], the previous study had an inclusion criterion of age 15 and above. This criterion led to the exclusion of some relevant published articles. Hence, DKA frequently occurs at the time of diabetes diagnosis, with younger children being at the highest risk, possibly because the symptoms of diabetes are more likely to go unnoticed in this age group [12,13]. Consequently, the current review is more comprehensive and inclusive. The insights from this meta-analysis could help identify effective interventions to reduce the severity and burden of DKA in Ethiopia.

## Materials and methods

### Data source and searching strategy

A thorough literature search was conducted using Google Scholar, PubMed/Medline, science direct and gray literature. The inclusion criteria covered all studies published in Ethiopia that explored diabetic ketoacidosis and its associated factors. We conducted a manual search for cross-references to discover additional pertinent articles. The Preferred Reporting Items for Systematic Reviews and Meta-Analyses (PRISMA) guidelines were employed for screening the articles [14]. The primary studies were searched across various databases by two authors, SFF and AMD. Mesh terms used and the number of databases retrieved from each database are illustrated in the annex part (**Annex I in S1 Annex**). The protocol has been registered on the PROSPERO-International Prospective Register of systematic reviews under registration number CRD42024553510.

### Study selection criteria

**Eligibility criteria.**   This comprehensive review comprises observational studies (including cross-sectional, case–control, cohort, and survey designs) reporting the prevalence of diabetic ketoacidosis in individuals with diabetes patients in Ethiopia, published between 2013 and 2023. The final search for this study was conducted on December 24, 2023. Our inclusion criteria encompassed articles published in the English language, and we considered studies conducted in both healthcare institutions and community settings. However, case reports, qualitative studies, studies with duplicate published literature and those lacking essential outcome data and articles lacking full text were excluded from this systematic review and meta-analysis.

**Data extraction.**   To obtain essential information from each selected paper, our review team developed a data extraction form using Microsoft Excel prior to the extraction process. Subsequently, two authors (SFF and AMD) independently extracted the required data from the included studies. The extracted data encompassed details such as the first author's name, publication year, study design, study setting, study area, participant sample size, total cases of diabetic ketoacidosis, associated factors, comparison group (Reference group), adjusted odds' ratio, and the lower and upper levels of the 95% confidence interval for the associated factors.

**Quality assessment for studies.**   The meta-analysis's reliability is contingent on the quality of the incorporated studies. Two authors (SFF and AMD) evaluated the risk of bias in the included studies using a modified version of the Newcastle–Ottawa Scale (NOS) designed for cross-sectional studies. Each item in the tool was completed for every study with a response of yes or no, and the study's quality was determined by totaling the scores assigned to each item, categorizing them as low bias ($\geq 7$ points), moderate bias (three to six points), and high bias ($\leq 3$ points). To enhance the review's validity, we specifically included studies with low and moderate risk of bias [15].

**Outcome measurement.**   The primary outcome was DKA, and it was defined within the framework of hyperglycemia (with blood glucose levels exceeding 200 mg/dL or 11 mmol/L) and the presence of the following criteria: a blood bicarbonate level below 15 mmol/L, a pH level below 7.30, a documented DKA diagnosis in medical records, or the presence of ketone bodies in the urine. The second outcome of this systematic review and meta-analysis reported the predictors of DKA among patients with DM.

**Data processing and analysis.**   Upon data extraction from all eligible studies, the information was entered, and data analysis was conducted using R software version 4.3.2. The random effects model was employed to estimate the overall pooled prevalence of diabetic ketoacidosis

and its primary components. Addressing heterogeneity among studies is crucial in meta-analysis. The I2 test statistic was utilized to assess the consistency of studies, examining the hypothesis that all included studies evaluated the same effect. As there was heterogeneity observed among the original studies (I2 = 100%, p < 0.001), a random effect model was deemed necessary. To accommodate between-study variance, a random effect meta-analysis was performed using the DerSimonian and Laird method for estimation. Additionally, potential publication bias was evaluated using funnel plots and Egger's tests, as recommended by various scholars. The results were presented through a forest plot (forest. meta).

**Publication bias and heterogeneity.** To address bias, comprehensive searches were carried out, and collaborative efforts among the authors were pivotal in maintaining objectivity and adhering to clear criteria during article selection, quality assessment, and data extraction. Qualitative scrutiny of the funnel plot ([16] package, the *funnel.meta*) graph was employed to explore potential publication bias [17]. Furthermore, Egger's correlation tests were applied at a 5% significance level to detect any presence of publication bias [18]. Subgroup analysis, considering study region, type of DM and study participants aimed to minimize random variations in primary study estimates. In the random-effect model, heterogeneity across studies was evaluated using inverse variance (I2) statistics and the corresponding p-value.

## Result

### Description of eligible studies

To provide the results of this review, we adhered to the PRISMA guidelines. Using PubMed, Google Scholar, Science Direct, and gray literature, we extracted 189 papers about Diabetic ketoacidosis. 116 papers were examined after duplicates were removed, and after reading the titles and abstracts of 54 of those articles were discarded. According to the research selection procedure chart, 23 studies were eligible and included in the final analysis (**Fig 1**).

### Magnitude of diabetic ketoacidosis among diabetic patents in Ethiopia

In this meta-analysis, from a total of 22 studies among 5722 study participants, the pooled prevalence of diabetic ketoacidosis among diabetic patients in Ethiopia is presented on a forest plot. Therefore, the pooled estimated prevalence of diabetic ketoacidosis among diabetic patients in Ethiopia was 46% (95% CI; 36, 57; I2 = 100%, P≤0.001) (**Fig 2**).

### Publication bias

The evaluation of publication bias involved examining the funnel plot visually and conducting the Egger's regression test. The funnel plot displayed asymmetry (**Fig 3**), and Egger's test yielded a p-value of 0.0161, indicating the presence of publication bias as suggested by both the plot and statistical significance. This bias could arise from the incorporation of diverse articles featuring distinct study designs, study populations, study durations, and assessment areas for factors associated with diabetic ketoacidosis. Subgroup analysis was conducted to delve deeper into the sources of heterogeneity, and the findings were subsequently presented. Trim and fill analysis was conducted to overcome the publication bias as shown in the funnel plot (**Fig 4**). Following trim and fill analysis, the pooled prevalence of DKA among DM patients in Ethiopia was 81.79% (62.95%, 100%).

### Subgroup analysis

We also performed subgroup analysis to explore the variations in diabetic ketoacidosis prevalence among different studies, considering factors such as study area, type of DM and

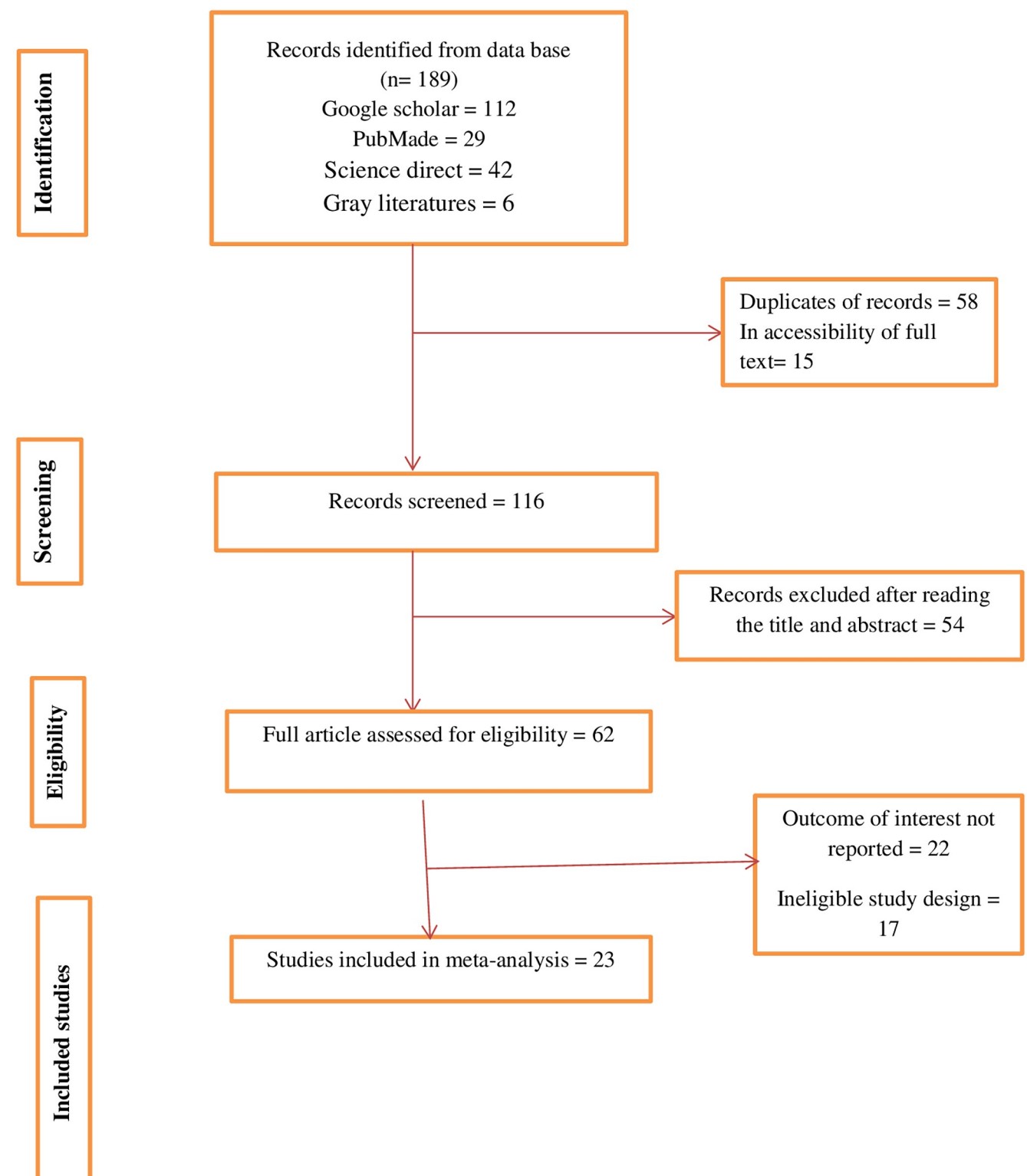

**Fig 1. PRISMA flow diagram describing the selection of studies included in the systematic review and meta-analysis of prevalence and determinants of diabetic ketoacidosis in Ethiopia.**

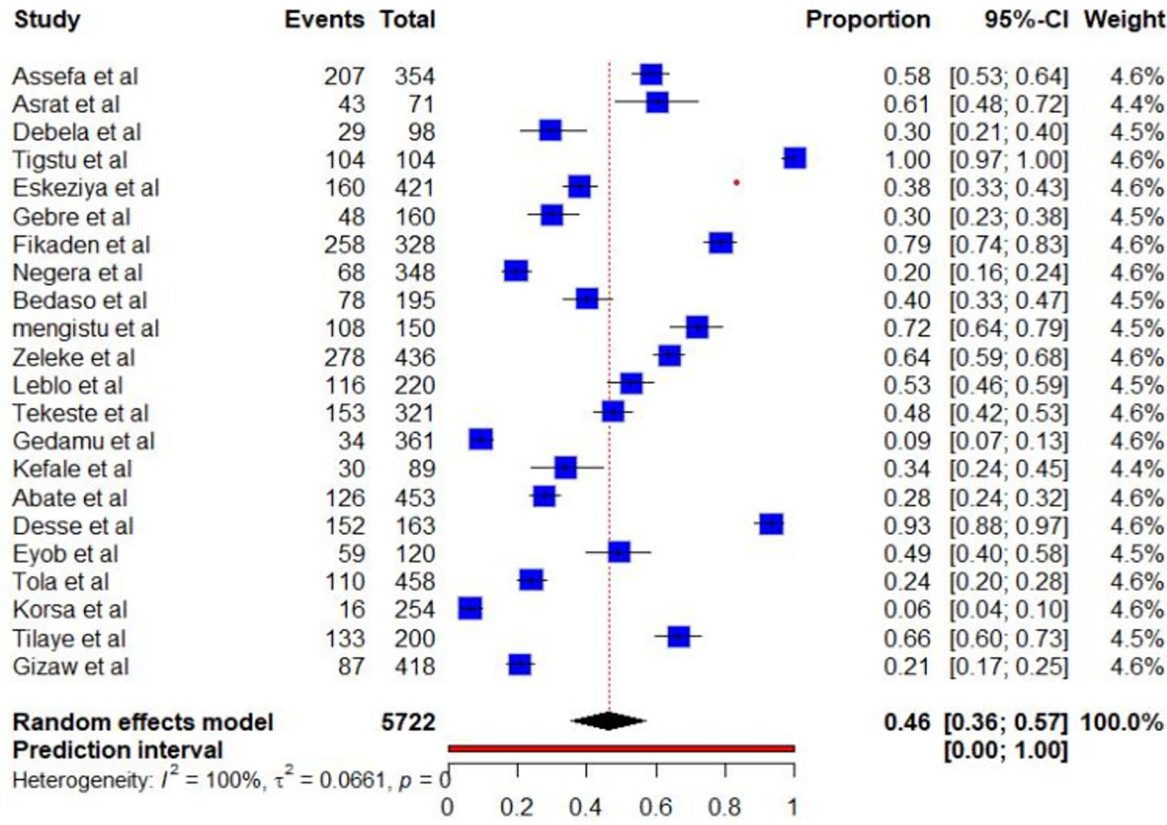

**Fig 2. Forest plot of the pooled prevalence of diabetic ketoacidosis among diabetic patients in Ethiopia.**

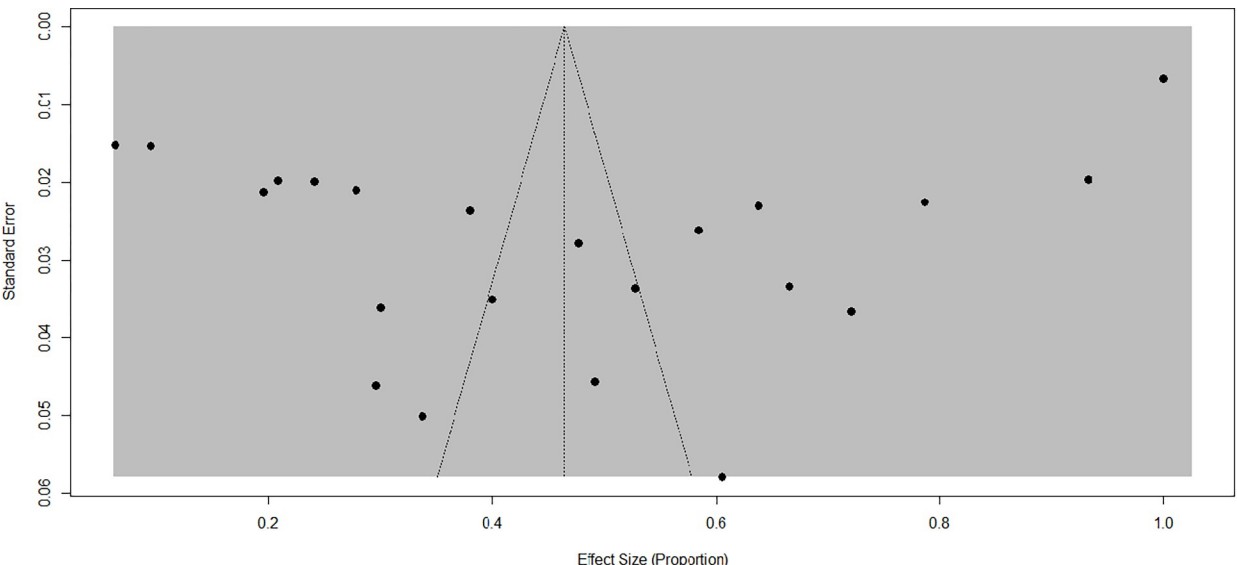

**Fig 3. Funnel plot depicts publication bias of included studies on prevalence of diabetic ketoacidosis in Ethiopia.**

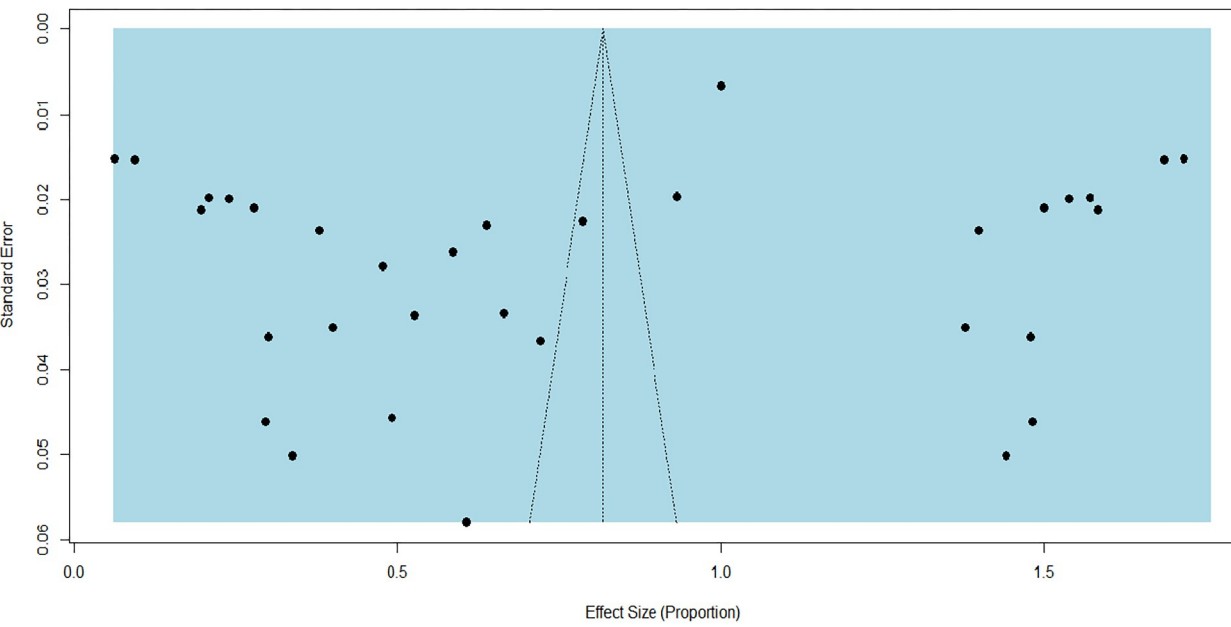

**Fig 4. Funnel plot after trim and fill analysis.**

participants (**Figs 5–7**). In the subgroup analysis based on study area, Amhara region exhibited the highest aggregated prevalence of diabetic ketoacidosis at 52.44% (95% CI: 35.76%, 69.12%), followed by the Oromia region with 49.83% (95% CI: 26.03%, 73.63%). Conversely, the lowest prevalence was observed in Addis Ababa at 32.52% (95% CI: 12.05%, 53.0%). The subgroup analysis focusing on study participants revealed that children had the highest combined prevalence of diabetic ketoacidosis, amounting to 68.20% (95% CI: 59.32%, 77.10%). In addition, the pooled prevalence of DKA is high among type I DM patients at 70% (95% CI: 61%, 77%) (**Annex II, Table 2 in S1 Annex**).

## Sensitivity analysis

Most of the effect size estimates, when excluding individual studies, seem to cluster around the overall effect size. There are a few studies (e.g., "Tigstu et al." and "Korsa et al.") whose exclusion results in effect size estimates that deviate more noticeably from the overall effect size, suggesting they may have a greater influence on the pooled result. The overall effect size appears relatively robust as no single study drastically changes the pooled effect size estimate, though there is some variability (**Fig 8**). Alternatively, it is shown in the table (**Table 1**).

## Baujat plot

A Baujat plot is useful for identifying studies that are particularly influential on the overall meta-analysis results. As shown in **Fig 9**, Tigstu et al. has the highest influence on the overall result and contributes significantly to heterogeneity, indicating a strong impact on the meta-analysis findings. Desse et al., Korsa et al., Gedamu et al., and Fikaden et al. show moderate influence and heterogeneity contributions, suggesting they are somewhat influential. Most studies are clustered in the bottom-left corner, indicating minimal influence on both the overall effect size and heterogeneity.

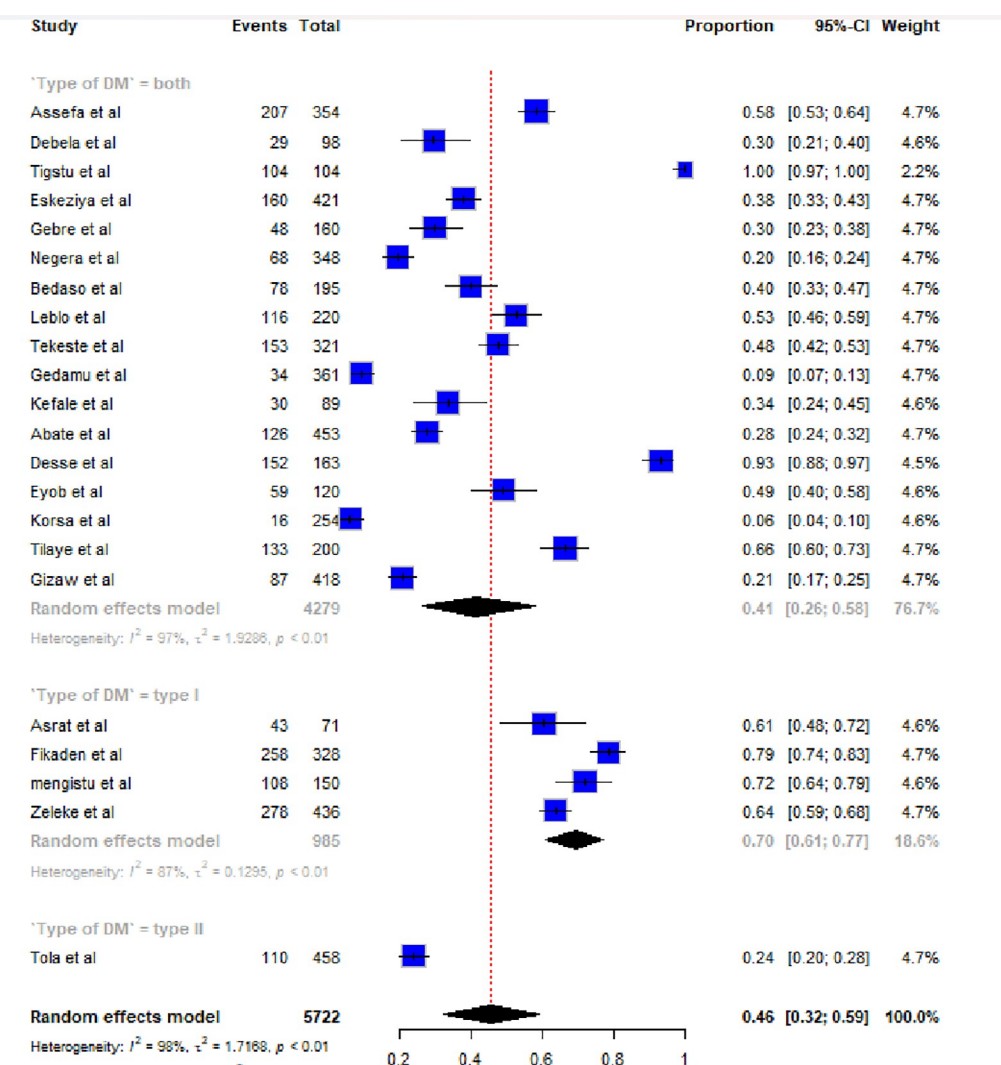

**Fig 5. Forest plot of pooled prevalence of diabetic ketoacidosis among diabetic patients in Ethiopia by type of DM.**

## Factors associated with diabetic ketoacidosis

In the meta-analysis, fixed-effects models were used to assess the association between medication discontinuations, presence of comorbidity, and presence of infection with diabetic ketoacidosis. The Egger test p-values for medication discontinuations, presence of comorbidity, and presence of infection were 0.17, 0.110, and 0.89, respectively, suggesting no evidence of publication bias. There was no significant heterogeneity observed for any of the predictors ($I^2 = 0\%$), and individuals with medication discontinuations have a 1.30 times higher odds of diabetic ketoacidosis compared to those without discontinuations (AOR = 1.30, 95 CI 1.20, 1.64). Individuals with comorbidities have a 1.53 times higher odds of diabetic ketoacidosis compared to those without comorbidities (AOR = 1.53, 95 CI 1.10, 2.20). Individuals with infections have, on average, 1.62 times higher odds of developing diabetic ketoacidosis compared to those without infections (AOR = 1.62, 95 CI 1.31, 1.98) (Table 2).

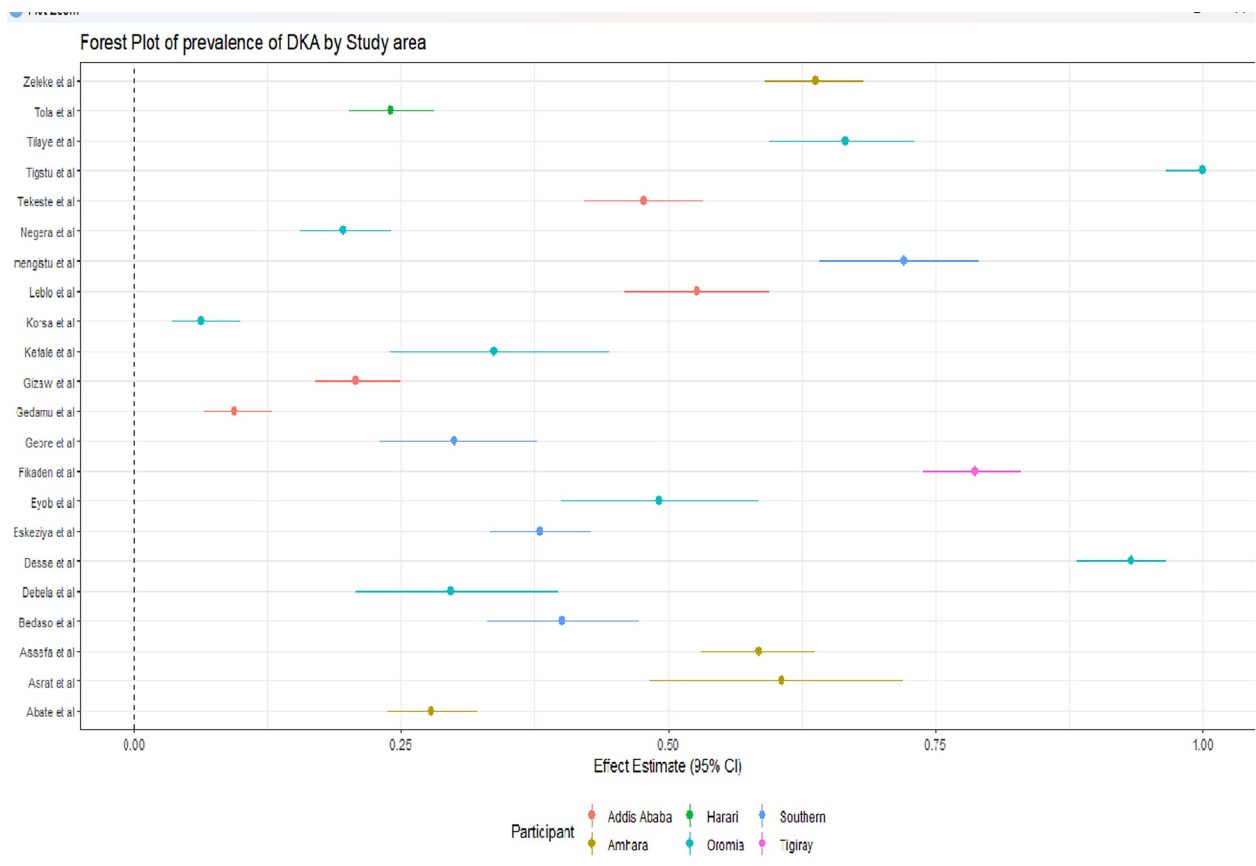

**Fig 6. Forest plot of the pooled prevalence of diabetic ketoacidosis in Ethiopia by study area.**

## Discussion

Acute complications of diabetes can arise relatively quickly and often require immediate attention. The two main acute complications associated with diabetes are diabetic ketoacidosis (DKA) and hyperglycemic hyperosmolar state (HHS). DKA typically occurs in individuals with type 1 diabetes but can also affect those with type 2 diabetes, especially in situations of severe insulin deficiency.

This systematic review and meta-analysis aim to provide updated aggregated estimates of diabetic ketoacidosis prevalence among individuals with diabetes in Ethiopia. The findings will offer valuable insights for policymakers, health planners, and the community.

The combined estimated prevalence of diabetic ketoacidosis (DKA) among diabetic patients in Ethiopia was 46% (95% CI: 36, 57; $I^2$ = 100%, P≤0.001), which is higher than the previously reported pooled prevalence of DKA in Ethiopia [11]. Howe ever, a lower prevalence has been reported in previous studies conducted at Cambridge University (18–22%) [19], Nigeria (12.2%) [20], and England (22.5–23.9%). The variation may be attributed to socio-cultural differences in health-seeking behavior, changes in diet and overall lifestyle resulting from increasing urbanization and economic development in the region.

Another retrospective study conducted at a Benin teaching hospital found the frequency of DKA in newly diagnosed adults to be 77.1% [21]. This higher frequency, compared to the present study, can be attributed to the inclusion of adults aged 15–35 years and a study period of 15 years. Our finding is in line with retrospective data from 2013 to 2014 that reported a 40% prevalence of DKA in Iraq [22].

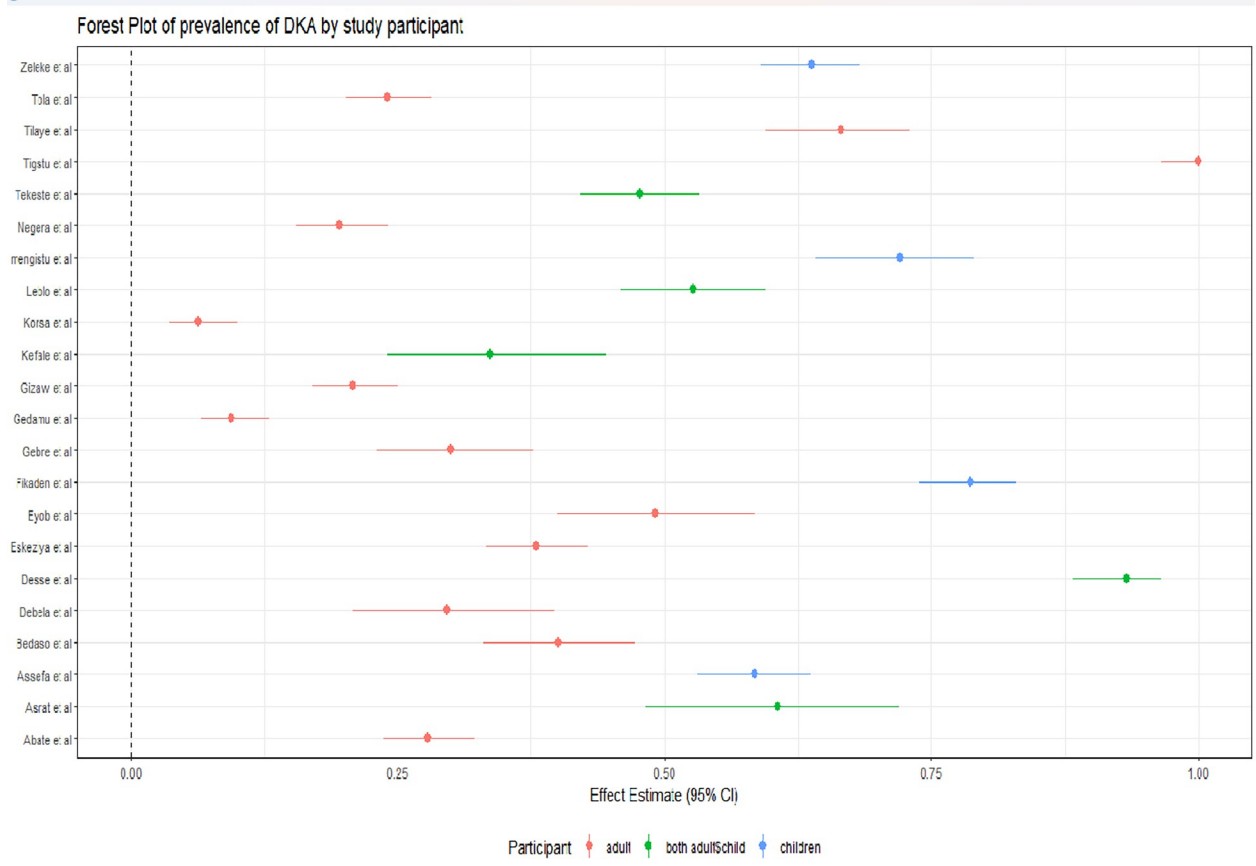

**Fig 7. Forest plot of the pooled prevalence of diabetic ketoacidosis in Ethiopia by study participants.**

The magnitude of diabetic ketoacidosis was higher than a study conducted in Kenya [23]. This might be a difference in sample size and populations. But it is comparable to a study conducted in China [24].

Our findings is higher than a study conducted in Europe, Australia, New Zealand, and the United States, the general prevalence of diabetic ketoacidosis (DKA) was 29.9% [16], varying from 19.5% in Sweden to 43.8% in Luxembourg. The mean prevalence of DKA at the time of type 1 diabetes (T1D) diagnosis was estimated to be 29.9%. Notably, in the United States, the prevalence increased from 35.3% to 40.6% between 2010 and 2016, reflecting a 2% annual rise in DKA occurrence at or near the diagnosis of type 1 diabetes [25].

There could be several possible justifications for the pooled prevalence of diabetic ketoacidosis (DKA) in Ethiopia being higher than that reported in studies conducted in Europe, Australia, New Zealand, and the United States. Here are some potential explanations. Ethiopia may face challenges in healthcare infrastructure and access compared to more developed regions. Limited access to medical care and diabetes management resources could contribute to delayed diagnosis and increased prevalence of DKA [26,27].

In addition, socioeconomic factors, including poverty and limited resources, may affect the ability of individuals in Ethiopia to afford and access diabetes care. Economic constraints can contribute to suboptimal diabetes management. Finally, late diagnosis of diabetes and limited awareness of the importance of early intervention may result in a higher proportion of individuals presenting with severe complications like DKA in Ethiopia [28,29].

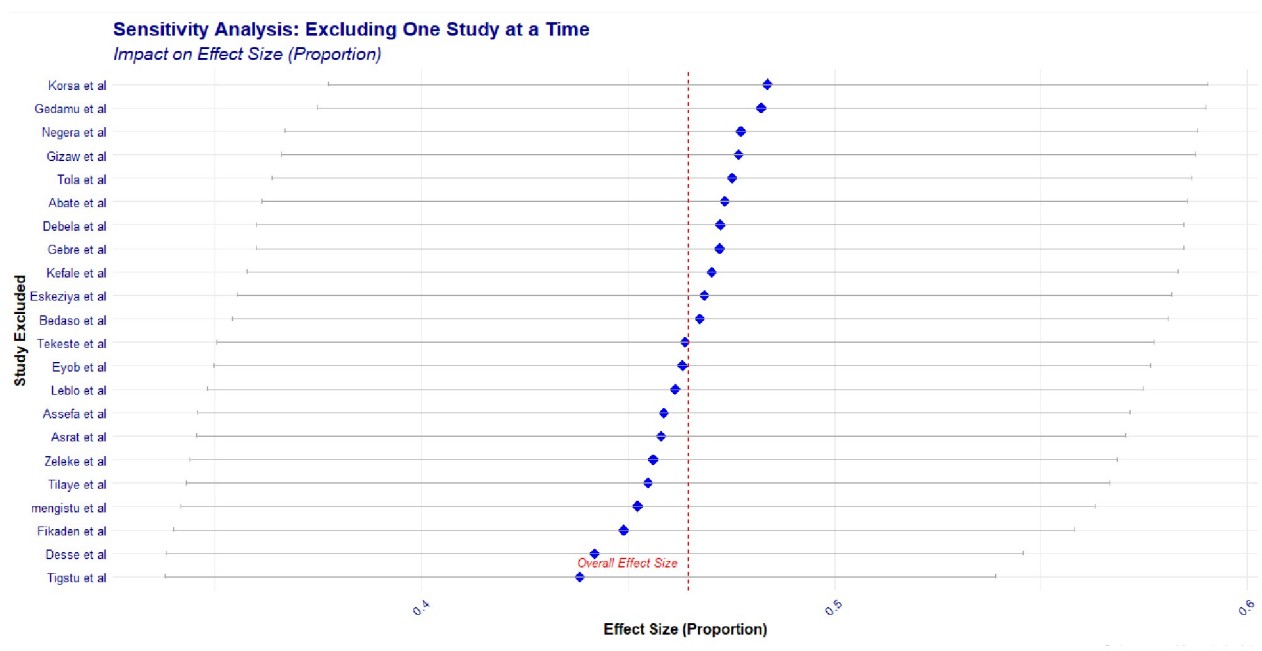

**Fig 8. Sensitivity analysis of forest plot of pooled prevalence of DKA among DM patients in Ethiopia.**

**Table 1. Leave-one-out sensitivity analysis results with confidence intervals.**

| Authors | Estimate | Lower CI | Upper CI |
|---|---|---|---|
| Assefa et al omitted | 0.4587012 | 0.3457906 | 0.5716118 |
| Asrat et al omitted | 0.4579652 | 0.3454207 | 0.5705098 |
| Debela et al omitted | 0.4723430 | 0.3600786 | 0.5846074 |
| Tigstu et al omitted | 0.4384704 | 0.3379782 | 0.5389626 |
| Eskeziya et al omitted | 0.4685007 | 0.3552705 | 0.5817308 |
| Gebre et al omitted | 0.4722467 | 0.3598930 | 0.5846005 |
| Fikaden et al omitted | 0.4489924 | 0.3399282 | 0.5580565 |
| Negera et al omitted | 0.4773716 | 0.3669344 | 0.5878088 |
| Bedaso et al omitted | 0.4675138 | 0.3542062 | 0.5808214 |
| mengistu et al omitted | 0.4523513 | 0.3416020 | 0.5631006 |
| Zeleke et al omitted | 0.4561511 | 0.3438916 | 0.5684106 |
| Leblo et al omitted | 0.4614710 | 0.3481486 | 0.5747934 |
| Tekeste et al omitted | 0.4638742 | 0.3503669 | 0.5773814 |
| Gedamu et al omitted | 0.4823001 | 0.3747390 | 0.5898611 |
| Kefale et al omitted | 0.4703842 | 0.3576473 | 0.5831211 |
| Abate et al omitted | 0.4733997 | 0.3613368 | 0.5854625 |
| Desse et al omitted | 0.4419132 | 0.3380756 | 0.5457508 |
| Eyob et al omitted | 0.4631809 | 0.3497957 | 0.5765662 |
| Tola et al omitted | 0.4752305 | 0.3638369 | 0.5866240 |
| Korsa et al omitted | 0.4838084 | 0.3773368 | 0.5902800 |
| Tilaye et al omitted | 0.4549228 | 0.3431156 | 0.5667299 |
| Gizaw et al omitted | 0.4767713 | 0.3660444 | 0.5874982 |

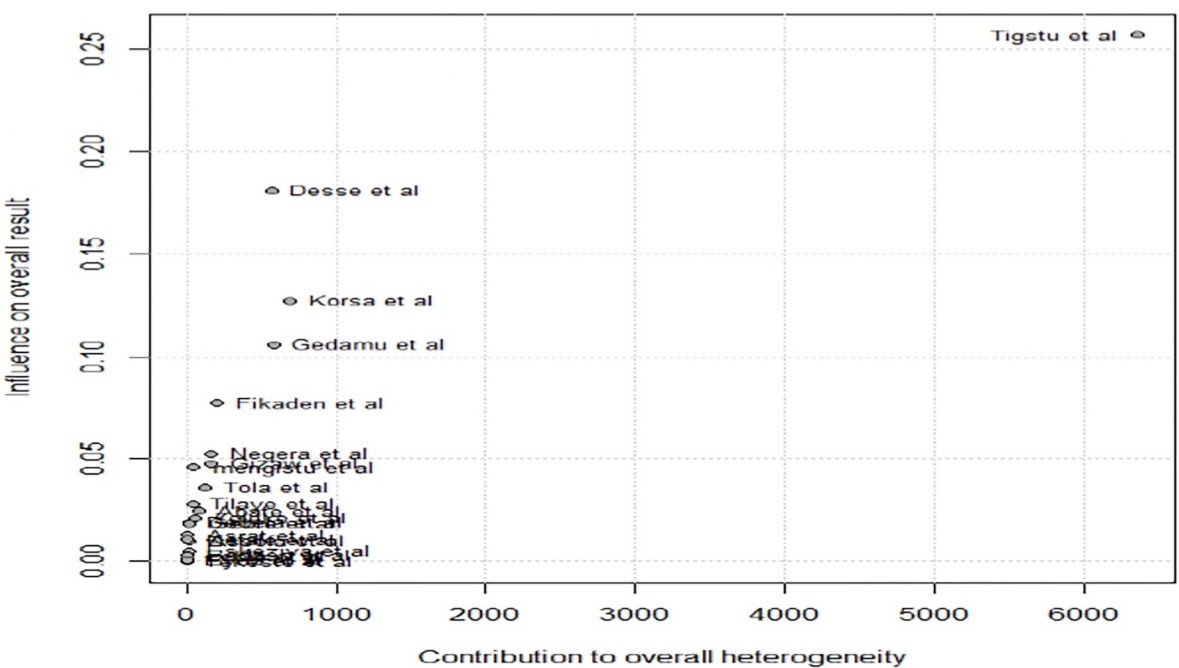

**Fig 9. Baujat plot of pooled prevalence of DKA among DM patients in Ethiopia.**

Our findings are comparable to those of sixty-five cohort studies involving over 29,000 children across 31 countries. The frequency of DKA at diagnosis ranged from 12.8% to 80%, with the highest frequencies observed in the United Arab Emirates, Saudi Arabia, and Romania, and the lowest in Sweden, the Slovak Republic, and Canada [30].

The findings of this study revealed factors that predict the occurrence of diabetic ketoacidosis in individuals with diabetes mellitus. Factors such as medication discontinuation, the presence of comorbidities, and the existence of infections were identified as significant determinants of diabetic ketoacidosis among patients with diabetes mellitus.

Individuals who stopped taking their medication had a 1.30 times higher likelihood of experiencing diabetic ketoacidosis compared to those who adhered to their medication regimen. This finding agreed with studies conducted in Mayo Hospital [31] and USA [32]. In addition, this finding was comparable with the studies conducted in a tertiary care hospital of Dhaka, Bangladesh, and the Soroka University Medical Center of Israel [33–35]. This might be discontinuing medications like oral hypoglycemic agents and insulin can elevate blood glucose levels by affecting glucose uptake, absorption, gluconeogenesis, glycogenolysis, and causing diabetic ketoacidosis through lipolysis or medications often suppress the processes of gluconeogenesis (glucose production by the liver) and glycogenolysis (breakdown of glycogen into glucose). Discontinuation leads to an increase in these processes, raising blood glucose levels

**Table 2. Summary of associated risk factors diabetic ketoacidosis among diabetic patients in Ethiopia.**

| Variables | Model | Egger test (P-value) | Status of heterogeneity | AOR (95% CI) | $I^2$ |
|---|---|---|---|---|---|
| Medication discontinuations | Fixed | 0.17 | No heterogeneity | 1.30 (1.20, 1.64) | 0 (0.0%, 84.7%) |
| Presence of comorbidity | Fixed | 0.11 | No heterogeneity | 1.53 (1.10, 2.20) | 0 (0.0%, 78.2%) |
| Presence of infection | Fixed | 0.89 | No heterogeneity | 1.62 (1.31, 1.98) | 0(0.0%, 82.4%) |

[36]. Therefore, it can be concluded that poor adherence to treatment is a significant contributing factor to DKA, highlighting the need for regular adherence assessments, particularly during patient reviews in diabetes clinics.

The existence of comorbidities was identified as a contributing factor to diabetic ketoacidosis, a finding supported by research conducted in the United States, Kenya, and Mayo Hospital [31,32] and multicenter analysis conducted in four countries (Germany, Australia, Switzerland, and Luxemburg) [37]. This might be comorbidities such as hypertension, chronic heart failure, and renal disease can directly affect insulin function. Insulin plays a crucial role in glucose regulation, and any disruption in its function that can lead to increased blood glucose levels, predisposing individuals to DKA [38].

Presence of infection is one, the predominant factors for diabetic ketoacidosis which, is consistent with findings in South Africa, Saudi Arabia and Colombia [39–41]. This might be that infections can trigger an inflammatory response, leading to increased insulin resistance. Insulin becomes less effective in facilitating glucose uptake by cells, resulting in elevated blood glucose levels. In response to infection, the body releases stress hormones, such as cortisol and catecholamines. These hormones can stimulate the liver to produce more glucose and interfere with insulin action, contributing to hyperglycemia. Hence, it's crucial for individuals with diabetes to manage infections promptly, monitor blood glucose levels closely, and adjust insulin doses as needed during illness to prevent the development of diabetic ketoacidosis. Regular communication with healthcare providers is essential to ensure appropriate management in the presence of infections [42–44].

## Limitations of the study

A number of the studies included in this review had a relatively small sample size, which may decrease the power of the study. This review has been limited to articles published only in English that cause reporting bias. Data was not found in all regions of the country. This can cause representative problems.

## Conclusions

The pooled prevalence of diabetic ketoacidosis among diabetic patients is remarkably high in Ethiopia. Medication discontinuations, presence of comorbidity, and presence of infection were independent determinants of diabetic ketoacidosis among diabetic patients. Hence, targeted initiatives should be implemented to improve medication adherence among diabetic patients and establish comprehensive diabetes management programs that address not only glycemic control but also comorbid conditions, including managing infections in diabetic patients.

## Supporting information

**S1 Table. A table showing the completed risk of bias and quality/certainty assessments for each study or outcome.**
(DOCX)

**S2 Table. List of all studies identified in the literature search, including those that were excluded from the analyses.**
(DOCX)

**S3 Table. Data extracted from each study for the reported systematic review and/or meta-analysis.**
(XLSX)

**S4 Table. PRISMA checklist.**
(DOCX)

**S1 Annex.**
(DOCX)

## Acknowledgments

The authors of the primary study are acknowledged.

## Author Contributions

**Conceptualization:** Sefineh Fenta Feleke, Anteneh Mengist Dessie, Zenebe Daniel Getachew, Fasikaw Kebede Bizuneh, Atitegeb Abera Kidie, Berihun Mulu Yayeh, Birtukan Gizachew Ayal, Natnael Amare Tesfa.

**Formal analysis:** Sefineh Fenta Feleke, Fasikaw Kebede Bizuneh, Berihun Mulu Yayeh, Birtukan Gizachew Ayal.

**Methodology:** Sefineh Fenta Feleke, Anteneh Mengist Dessie, Atitegeb Abera Kidie, Berihun Mulu Yayeh, Birtukan Gizachew Ayal, Natnael Amare Tesfa.

**Software:** Sefineh Fenta Feleke, Anteneh Mengist Dessie, Zenebe Daniel Getachew, Fasikaw Kebede Bizuneh, Atitegeb Abera Kidie, Berihun Mulu Yayeh, Birtukan Gizachew Ayal, Natnael Amare Tesfa.

**Validation:** Sefineh Fenta Feleke, Natnael Amare Tesfa.

**Visualization:** Fasikaw Kebede Bizuneh, Atitegeb Abera Kidie, Natnael Amare Tesfa.

**Writing – original draft:** Sefineh Fenta Feleke, Anteneh Mengist Dessie, Zenebe Daniel Getachew.

**Writing – review & editing:** Sefineh Fenta Feleke, Zenebe Daniel Getachew.

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
