## [Decision Letter · Decision Letter 0]

6 May 2024

PONE-D-24-05222Burden of diabetic ketoacidosis and its predictors among diabetic patients in Ethiopia: Systematic review and meta-analysisPLOS ONE

Dear Dr. Feleke,

Thank you for submitting your manuscript to PLOS ONE. After careful consideration, we feel that it has merit but does not fully meet PLOS ONE’s publication criteria as it currently stands. Therefore, we invite you to submit a revised version of the manuscript that addresses the points raised during the review process.

We look forward to receiving your revised manuscript.

Kind regards,

Felix Bongomin, MB ChB, MSc, MMed, FECMM

Academic Editor

PLOS ONE

Journal Requirements:

2. We note that your Data Availability Statement is currently as follows: All relevant data are within the manuscript and its Supporting Information files

3. Please ensure that you refer to Figure 2, 5 and 6 in your text as, if accepted, production will need this reference to link the reader to the figure.

**Additional Editor Comments:**

These work needs major revision and re-write up following the advise of the reviewers .

Also , there should be evidence of registration on PROSPERO before we progress further with this manuscript.

Reviewers' comments:

Reviewer's Responses to Questions

**Comments to the Author**

1. Is the manuscript technically sound, and do the data support the conclusions?

Reviewer #1: No

Reviewer #2: Yes

2. Has the statistical analysis been performed appropriately and rigorously? 

Reviewer #1: Yes

Reviewer #2: Yes

3. Have the authors made all data underlying the findings in their manuscript fully available?

Reviewer #1: No

Reviewer #2: No

4. Is the manuscript presented in an intelligible fashion and written in standard English?

Reviewer #1: No

Reviewer #2: No

5. Review Comments to the Author

Reviewer #1: 1. While it is not mandatory to register the systematic review on PROSPERO, it is highly recommended to do so.

2. Regarding the searching strategies, I suggest placing all MeSH terms in an appendix instead of listing them within the main manuscript. Additionally, provide a table format listing all the searching databases along with their respective numbers of articles found. It's worth noting that the current searching databases may not be sufficient to capture all relevant literature, potentially leading to missed papers due to inadequate search strategies.

3. In terms of eligibility criteria, could you clarify whether the authors excluded studies with duplicate published literature and those lacking essential outcome data? This clarification is crucial to address potential duplication. Additionally, you've listed observational studies reporting the prevalence of diabetic ketoacidosis in individuals with diabetes patients in Ethiopia between 2013 and 2023. However, could you justify the exclusion of the following articles: Getie 2021, Asfaw 2021, Eyob Tediso 2023, Korsa 2019, Tilaye 2021, Abate 2023, Kefale 2016, Desse 2015, Gizaw 2015, and Tola 2021?

4. In the Outcome Measurements section, please elaborate on the primary and secondary outcomes of the study.

5. In the Statistical Analysis section, considering you reported associations using adjusted odds ratios, it is recommended to report the extraction of AOR with their upper and lower bounds in the analysis section. Could you also provide the I2 ranges to assess heterogeneity and include a reference? Subgroup analysis should include the type of diabetes mellitus as it is crucial for diabetic ketoacidosis prevalence. Additionally, please mention the performance of sensitivity analysis to examine the influence of a single study on the overall estimate under the analysis section.

6. In the Results section:

• Keep the same searching databases: Google Scholar, PubMed/Medline, Web of Sciences, and grey literature.

• Regarding Figure 1, where "53 articles were excluded," include the issue of duplication under the eligibility criteria.

• The overall number of participants should be provided when assessing the pooled prevalence (e.g., From 14 studies with ‘__’ study participants, the pooled prevalence of DKA among DM patients in Ethiopia was…).

7. The Discussion should commence by highlighting the main findings (the pooled prevalence and associated factors) before discussing them separately. It would be beneficial if the authors compared the findings with those from high-income countries for coherence."

Overall, it appears that several important aspects have been overlooked by the authors, particularly the exclusion of prominent articles essential for conducting a comprehensive systematic review.

Reviewer #2: This manuscript presents the burden of diabetic ketoacidosis and its predictors among diabetic patients in Ethiopia. The authors conducted comprehensive review and provided a detailed analysis of their findings. While the manuscript is well-written and generally clear, there are some minor issues to address:

General comments

1. Rewrite the methods section of the abstract.

2. There is continuous repetition of ideas in the last paragraph-h of the introduction please reexamine and correct it.

3. In you document you used the phrase “diabetic patents instead of “diabetic patients”.

4. There are also many typos errors through the manuscript or to be rephrase paragraphs please wisely address it.

5. In the methods, you should have complied with the PRISMA (Preferred Reporting Items for Systematic Reviews and Meta-analyses) guidelines. Why did you strictly follow the PRISMA 2020 Checklist and PRISMA 2020 flow diagram ? Since figure 1 is not similar to the PRISMA 2020 flow diagram, please visit this link: PRISMA (prisma-statement.org)

6. Please check the sensitivity analysis (the impact of each study on the pooled effect size).

7. State any a priori levels of significance and whether the tests were 1- or 2-sided.

8. Some of your discussions are ambitious; as such, they are not supported by sufficient research or study across the globe. It should be solved for the resubmitted or revised manuscript.

9. Where are other statistical results like Begg’s test?

10. There is repetition of ideas throughout the manuscript.

11. Did you think you searched the available articles extensively?

12. All abbreviations you use in your manuscript should be written in full the first time, and you should use the abbreviated form in other parts of your manuscript.

13. The references are from 2003 to 2021. It is better to update the old references if there are any currently conducted articles available across the globe

14. There are many more editorial, punctuation, and language problems. Please address these issues with English-language professionals.

15. Finally, please check all references to ensure that none of the cited articles have been retracted. You can use the Retraction Watch database, available here (http://retractiondatabase.org/)

Dear Authors, congratulations again for the great job.

6. PLOS authors have the option to publish the peer review history of their article (what does this mean?). If published, this will include your full peer review and any attached files.

Reviewer #1: **Yes: **Eyob Girma Abera

Reviewer #2: No

---

## [Author Response · Author response to Decision Letter 0]

11 Jun 2024

For assessing publication bias, Egger's test is generally considered better due to its higher sensitivity. Or, Begg’s is less sensitive and may fail to detect publication bias, especially when the number of studies is small or when there is mild to moderate bias.

---

## [Decision Letter · Decision Letter 1]

19 Jul 2024

PONE-D-24-05222R1Burden of diabetic ketoacidosis and its predictors among diabetic patients in Ethiopia: Systematic review and meta-analysisPLOS ONE

Dear Dr. Feleke,

Thank you for submitting your manuscript to PLOS ONE. After careful consideration, we feel that it has merit but does not fully meet PLOS ONE’s publication criteria as it currently stands. Therefore, we invite you to submit a revised version of the manuscript that addresses the points raised during the review process.

We look forward to receiving your revised manuscript.

Kind regards,

Felix Bongomin, MB ChB, MSc, MMed, FECMM

Academic Editor

PLOS ONE

Journal Requirements:

Reviewers' comments:

Reviewer's Responses to Questions

**Comments to the Author**

1. If the authors have adequately addressed your comments raised in a previous round of review and you feel that this manuscript is now acceptable for publication, you may indicate that here to bypass the “Comments to the Author” section, enter your conflict of interest statement in the “Confidential to Editor” section, and submit your "Accept" recommendation.

Reviewer #2: (No Response)

2. Is the manuscript technically sound, and do the data support the conclusions?

Reviewer #2: Yes

3. Has the statistical analysis been performed appropriately and rigorously? 

Reviewer #2: Yes

4. Have the authors made all data underlying the findings in their manuscript fully available?

Reviewer #2: Yes

5. Is the manuscript presented in an intelligible fashion and written in standard English?

Reviewer #2: No

6. Review Comments to the Author

Reviewer #2: General comments

1. Dear authors, as far as I know one work found below is existed regarding DKA. Therefore, at the end paragraph of your introduction please state the gabs of the previously published work and state what you fill in current work. Abera, E. G., Yesho, D. H., Erega, F. T., Adulo, Z. A., Gebreselasse, M. Z., & Gebremichael, E. H. (2024). Burden of diabetic ketoacidosis among patients with diabetes mellitus in Ethiopia: a systematic review and meta-analysis. BMJ open, 14(2), e077151. https://doi.org/10.1136/bmjopen-2023-077151

2. The databases mentioned in the abstract (PubMed/MEDLINE, Google Scholar, and gray literature) and method (Google Scholar, PubMed/Medline, Web of sciences, and grey literature) section varies. How it happen?

3. Please perform sensitivity analysis

4. In the discussion section, your study did not give credit for African and Asian country studies. Also, your study result is not well discussed with the previous systematic review and meta-analysis studies conducted in Ethiopia and abroad.

5. In your study, the justifications written in the discussion sections of all paragraphs are not supported by previously published/conducted studies.

6. Please check the language again.

7. PLOS authors have the option to publish the peer review history of their article (what does this mean?). If published, this will include your full peer review and any attached files.

Reviewer #2: No

---

## [Author Response · Author response to Decision Letter 1]

26 Jul 2024

Reviewers comments Authors response 

Reviewer 2 

1. Dear authors, as far as I know one work found below is existed regarding DKA. Therefore, at the end paragraph of your introduction please state the gabs of the previously published work and state what you fill in current work. Abera, E. G., Yesho, D. H., Erega, F. T., Adulo, Z. A., Gebreselasse, M. Z., & Gebremichael, E. H. (2024). Burden of diabetic ketoacidosis among patients with diabetes mellitus in Ethiopia: a systematic review and meta-analysis. BMJ open, 14(2), e077151. https://doi.org/10.1136/bmjopen-2023-077151

Thank you very much for your insight full comments 

We made a revision according to it. 

2. The databases mentioned in the abstract (PubMed/MEDLINE, Google Scholar, and gray literature) and method (Google Scholar, PubMed/Medline, Web of sciences, and grey literature) section varies. How it happen? Thank you very much.

We made a revision on it.

3. Please perform sensitivity analysis Thank you for your insight 

We made it. 

4. In the discussion section, your study did not give credit for African and Asian country studies. Also, your study result is not well discussed with the previous systematic review and meta-analysis studies conducted in Ethiopia and abroad. Thanks for your constructive comments:

We made a revision on it. 

5. In your study, the justifications written in the discussion sections of all paragraphs are not supported by previously published/conducted studies. We would like to acknowledge for your deep insight: We made a revision on it.

6. Please check the language again. We would like to acknowledge for your deep insight : We made a revision on it.

Editor’s comment 

1. Please review your reference list to ensure that it is complete and correct. If you have cited papers that have been retracted, please include the rationale for doing so in the manuscript text, or remove these references and replace them with relevant current references. Any changes to the reference list should be mentioned in the rebuttal letter that accompanies your revised manuscript. If you need to cite a retracted article, indicate the article’s retracted status in the References list and also include a citation and full reference for the retraction notice We made a revision on it.

---

## [Editor Report · Decision Letter 2]

6 Aug 2024

Burden of diabetic ketoacidosis and its predictors among diabetic patients in Ethiopia: Systematic review and meta-analysis

PONE-D-24-05222R2

Dear Dr. Feleke,

We’re pleased to inform you that your manuscript has been judged scientifically suitable for publication and will be formally accepted for publication once it meets all outstanding technical requirements.

Kind regards,

Felix Bongomin, MB ChB, MSc, MMed, FECMM

Academic Editor

PLOS ONE
---

## [Editor Report · Acceptance letter]

2 Sep 2024

PONE-D-24-05222R2 

PLOS ONE

Dear Dr. Feleke, 

I'm pleased to inform you that your manuscript has been deemed suitable for publication in PLOS ONE. Congratulations! Your manuscript is now being handed over to our production team.

Kind regards, 

on behalf of

Dr. Felix Bongomin 

Academic Editor

PLOS ONE